# Modality-Specific Strategies for Medical Image Segmentation using Lightweight SAM Architectures

Thuy Dao*[1][0000−0003−0715−3555], Xincheng Ye*[1][0000−0002−2544−835X], Joshua Scarsbrook[1][0000−0002−0071−5466], Gowrienanthan Balarupan[1][0009−0002−3563−2887], Fernanda L. Ribeiro**[1][0000−0002−1620−4193], and Steffen Bollmann**[1,2][0000−0002−2909−0906]

[1] School of Electrical Engineering and Computer Science, University of Queensland, Australia
[2] Queensland Digital Health Centre, University of Queensland, Australia
`s.bollmann@uq.edu.au`

**Abstract.** Medical image segmentation tasks are often intricate and require medical domain expertise. Recent advancements in deep learning have expedited these demanding tasks, transitioning from specialized models tailored to each task to versatile foundation models capable of accommodating various image modalities. However, many of these foundation models are optimized for GPU computation, necessitating significant computational resources and constraining their practical utility in clinical settings. Furthermore, their variable accuracy across modalities and novel domains undermines their reliability in clinical practice. To address these limitations, we undertake a comparative investigation into deploying medical image segmentation models on CPU, focusing on accuracy and runtime efficiency, as part of the "*CVPR 2024: Segment Anything In Medical Images On Laptop*" challenge. Our methodology employs different models customized for each modality, including pre-trained EfficientViT-SAM and LiteMedSAM to yield the most precise and efficient outcomes. Additionally, to bolster model performance for datasets featuring small regions of interest, such as PET scans, we integrate a majority voting mechanism. We optimize runtime using the OpenVINO format within a C++ inference script. This approach improves inference runtime while maintaining competitive accuracy, achieving an average DSC score of **0.86** on the validation set and **0.75** on the testing set with an average runtime of **4.61s** on testing set. Notably, given that most modalities are evaluated in a zero-shot manner, our findings suggest that the zero-shot capability of foundation models can be further refined through dataset-specific inference strategies.

**Keywords:** Multi-modality · Zero-shot · OpenVINO · CPU Deployment

---

* These authors contributed equally to this paper.
** These authors contributed equally to this paper.

## 1   Introduction

Medical image segmentation, which involves manually delineating regions of interest (ROI), is a time-consuming endeavor. Furthermore, it requires a significant level of domain expertise for the precise identification of relevant landmarks and segmentation. The advent of deep learning marks a significant step forward to an automated solution for medical image segmentation [9]. Nevertheless, the variance between different medical image modalities and segmentation tasks makes developing a general segmentation model challenging [19]. However, recent advances in leveraging foundation models [1] have the potential to address these challenges. For example, by leveraging these pre-trained foundation models with prompt engineering techniques, one can optimize model performance without the need for model fine-tuning using large-scale medical image datasets.

Notably, the Segment Anything model (SAM) [5], a vision foundation model trained on a billion masks from 11 million natural scene images, has demonstrated immense potential to automate segmentation tasks. SAM is a promptable model—that is, it accepts prompts to guide segmentation, such as points, bounding boxes, or masks. While SAM shows strong zero-shot segmentation capabilities, differences in data statistics (intensity ranges and distributions) and inhomogeneity of medical images compared to natural scene images pose significant challenges, potentially limiting its performance [13,4,2].

Accordingly, MedSAM has been proposed to address the limited generalizability of SAM for medical image segmentation. It is a SAM-based model finetuned on one million medical image-ground truth segmentation pairs across 10 modalities [11]; it demonstrates a significant improvement in zero-shot medical image segmentation tasks in comparison to the original SAM. Nevertheless, given the large architecture footprint inherited from SAM, inference (or mask generation) requires GPU resources to perform efficiently and timely. This computing infrastructure requirement impedes the deployment of MedSAM in real-world scenarios, such as in clinical settings.

Therefore, there is a need for lightweight, promptable medical image segmentation models that can be deployed on laptops or edge devices without relying on expensive and scarce GPU resources. Accordingly, significant effort has been invested in various optimization techniques, such as model distillation, quantization, and pruning[12]. For example, LiteMedSAM consists of a distilled[15,8] version of MedSAM resulting in a more compact and efficient model, with reduced model size and, hence, inference time. Similarly, EfficientViT-SAM [18] replaces the heavy encoder of SAM with EfficientViT [7] through distillation. These studies have demonstrated the effectiveness of transferring knowledge to lightweight image encoders, showcasing their ability to reduce model size and runtime while maintaining accuracy to a considerable extent. However, their efficacy for zero-shot transfer to the medical imaging domain remains to be further investigated.

Here, we leverage these lightweight foundation models for efficient medical image segmentation using an edge device with limited memory and hardware (a 3.6GHz Intel CPU with 8G of RAM). To date, there is no one-size-fits-all solu-

tion for medical image segmentation; hence, we opt to use multiple lightweight foundation models in conjunction. Specifically, we explore the performance of different models—EfficientViT-SAM[18], EfficientSAM[16], and LiteMedSAM— across medical image modalities by comparing their accuracy and runtime. We further enhance model performance for 3D data with small regions of interest, such as PET scans, by integrating multiple-view knowledge and employing a majority voting strategy to combine segmentations across anatomical views. To reduce the model size, we converted the model into a lightweight format using OpenVINO. Finally, to accelerate deployment efficiency, we use a C++ inference script with an embedding caching mechanism that reduces runtime compared to the Python-based approach due to its compiled nature, optimized memory management, and direct hardware interaction [6]. This combination of strategies results in significant improvement in runtime while maintaining comparable accuracy on the validation set and testing set, demonstrating the potential of our solution to make advanced medical image segmentation models more accessible and efficient in practice without requiring an immense amount of labeled data for model fine-tuning.

## 2  Methods

### 2.1  Pre-processing

In the pre-processing phase, the intensity of each grey-scale 2D image (or 2D slices from 3D medical images) was normalized to the range of [0, 255]. Then, Gaussian normalization (which may produce negative values) was applied for EfficientSAM and EfficientViT-SAM, while min-max normalization (values are in the range of [0,1]) was applied for LiteMedSAM-based inference. The normalized images are then either padded or resized to match the required input dimension from each model (EfficientSAM: input image size of 1024×1024; EfficientViT-SAM: input image size of 512×512; LiteMedSAM: input image size of 256×256). Our models were validated using 3,278 images from the validation set (Table 1; see section 3.1 for more information about the data).

**Table 1.** The validation set consists of both 2D images and 3D volumes across various modalities. The 2D data includes images from CT, MR, Microscopy, Dermoscopy, Endoscopy, Fundus, X-Ray, and Ultrasound—US, while the 3D data comprises volumetric scans from CT, MR and PET.

| Modality | Microscopy | Dermoscopy | Endoscopy | Fundus | X-ray | US | PET | MR | CT |
|---|---|---|---|---|---|---|---|---|---|
| Number of Subjects | 50 | 66 | 200 | 10 | 581 | 600 | 3 | 628 | 1140 |

### 2.2  Proposed Method

Our approach (Figure 1) involves a tailored selection of SAM-based models for each medical image modality, combined with prompt engineering techniques

designed to enhance segmentation accuracy. By customizing the model and prompts based on the specific characteristics of each modality, we aim to optimize performance across diverse imaging types and improve the overall robustness of the segmentation process.

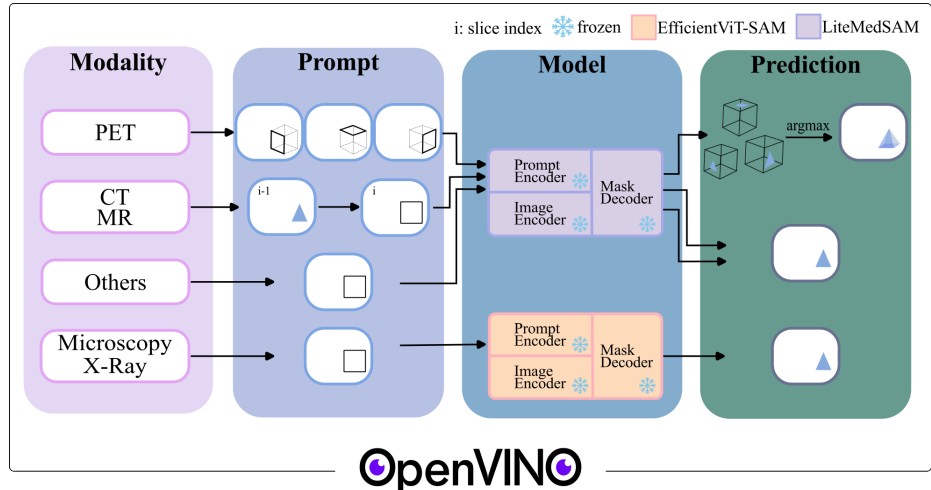

**Fig. 1.** Modality-specific strategy for medical image segmentation in edge device deployment.

**Model selection.** Our approach combines modality-specific strategies that leverage the potential of zero-shot generalizability of lightweight, pre-trained SAM-based foundation models. We found that EfficientViT-SAM[18], LiteMed-SAM, and EfficientSAM[16] demonstrated different generalizability across medical imaging modalities, requiring an empirical selection of inference models based on their performance.

**Prompt engineering.** To enhance the zero-shot performance of foundation models, one promising approach is to explore different model prompts [14]. Prompts—a click (point) or a bounding box—provide the spatial priors for target location and segmentation. Specifically for 3D medical data, an ROI within the current inference slice shares a similar spatial location with the same ROI in adjacent slices. Given that these foundation models only take 2D images as input, it is possible to use the mask prediction of the current inference slice to generate a bounding box for the next inference slice. This strategy can be leveraged to improve the prompt by narrowing the bounding box to include only the ROI in an automated fashion.

**3D data segmentation.** Another challenge in deploying foundation models is the potential lack of consideration for 3D spatial information. A single anatomical view (or plane) of 3D images, specifically the axial view, is commonly used for inference with 2D foundation models. However, some regions of interest may be easier to segment by considering other anatomical views. Here, we also leverage a majority voting mechanism to integrate segmentation from multiple views for PET image segmentation.

## 2.3   Post-processing

The model-generated mask underwent post-processing steps, which included cropping the padded space from the pre-processing step and resizing the mask to the original image dimensions using a linear interpolation algorithm. Subsequently, the output logits were thresholded at a value of 0.

## 2.4   Inference efficiency

**Model format.** Efficient deep learning model deployment is challenging due to dependencies on specific frameworks, libraries, and computational environments. Moreover, large amounts of model weights and intricate architectures make model deployment take an extensive time to run without GPU accelerators. Thus, we leveraged the Open Visual Inference and Neural Network Optimization (OpenVINO) project to enhance model efficiency. OpenVINO stores model graphs in a lightweight format and provides a C++ API optimized for Intel hardware, reducing initialization and runtime. We exported our model to the ONNX format, taking advantage of its graph optimization features, and then converted the ONNX graph to OpenVINO for execution with a C++ pipeline.

**Embedding cache.** Given that the image encoder is the most computationally expensive part of SAM-based models, we cache the image embedding for 3D inference with multiple ROIs to avoid recalculation.

**Docker image.** Regarding runtime evaluation, our results include Docker image loading time, which is significantly impacted by the Docker image size. We adjusted the base image to include only the operating system, system libraries, and necessary libraries for imaging processing and model inference as the initial layer for building a Docker image. This adjustment notably decreased loading time compared to the original Docker image provided by the CVPR challenge.

# 3   Experiments

## 3.1   Dataset and evaluation measures

The "CVPR 2024: Segment Anything In Medical Images On Laptop" challenge dataset consists of three subsets: 1) more than one million image-ground truth

segmentation pairs of pre-processed training data; 2) 3,278 image-bounding box pairs of pre-processed validation data; 3) 10 testing set demos from 10 different modalities with image-bounding box-ground truth segmentation triads. The accuracy and runtime were evaluated on the validation set and **final** testing set (as per the challenge).

We evaluated segmentation accuracy using two distinct metrics, Dice Similarity Coefficient (DSC) and Normalized Surface Dice (NSD), and the runtime as our deployment efficiency measure. Models were deployed using CPU restrained within 8GB of memory by Docker. These metrics collectively contribute to the ranking computation of the challenge.

### 3.2   Implementation details

**Environment settings.** Table 2 presents the development environment and general requirements.

**Table 2.** Development environments and requirements.

| System | Ubuntu 20.04.6 LTS |
|---|---|
| CPU | AMD EPYC-Milan Processor@2.6GHz |
| RAM | 120GB |
| GPU (number and type) | One NVIDIA A100 40GB |
| CUDA version | 12.0 |
| Programming language | Python 3.10 |
| Deep learning framework | torch 2.0.1, torchvision 0.15.2 |
| Specific dependencies | N/A |
| Code | https://github.com/NeuroDesk/cvpr-sam-on-laptop-2024 |

**Training protocols.** To improve the performance of LiteMedSAM for specific modalities, including PET and microscopy, we explored model fine-tuning using Sharpness-aware Minimization [3] for loss optimization. Sharpness-aware Minimization considers regions in the loss landscape with uniformly low values instead of solely focusing on achieving the lowest possible loss value. This strategy aims to improve the robustness of model performance given our small training set. However, given the memory-performance gain trade-off, we did not include the fine-tuned model in our final challenge submission.

Initially, we converted 3D PET images into 2D slices and augmented these alongside microscopy images using random flips. Considering training efficiency, we only fine-tuned the image encoder of LiteMedSAM, keeping the remaining parameters frozen. Detailed training protocols are listed in Table 3.

---

⋆ https://github.com/MrYxJ/calculate-flops.pytorch
⋆⋆ https://github.com/lfwa/carbontracker/

**Table 3.** Training protocols.

| | |
|---|---|
| Pre-trained Model | LiteMedSAM |
| Batch size | 16 |
| Patch size | 16×16×3 |
| Total epochs | 50 |
| Optimizer | AdamW [10], Sharpness-aware Minimization[3] |
| Initial learning rate (lr) | 0.00005 |
| Lr decay schedule | ReduceLROnPlateau |
| Training time | 20 hours |
| Loss function | DiceLoss + BCEWithLogitsLoss + MSELoss |
| Number of model parameters | 9.79M* |
| Number of flops | 147.57 GFLOPS* |
| $CO_2$eq | 1 Kg** |

## 4    Results and Discussion

The runtime and DSC scores were compared across nine modalities using LiteMed-SAM, EfficientSAM, and EfficientViT-SAM models on 450 images sampled from the training set (50 images for each modality), for which ground-truth masks were available. The results (Figure 2) indicate that LiteMedSAM demonstrated great performance while maintaining a competitive runtime for most modalities. However, for microscopy and X-Ray data, EfficientViT-SAM outperformed LiteMedSAM by 10.31% DSC and 6.12% DSC, respectively, even though LiteMed-SAM was trained using these data. Based on these findings, EfficientViT-SAM was selected for microscopy and X-Ray images, while LiteMedSAM was selected for the rest of the modalities.

As LiteMedSAM performed worse on PET images, we explored two methods to further improve segmentation accuracy: 1) fine-tuning LiteMedSAM with PET data; 2) using the pre-trained LiteMedSAM with a majority voting mechanism that incorporates 3D spatial information across segmentations generated from each anatomical view (axial, sagittal, and coronal). As shown in Figure 3, the majority voting mechanism using LiteMedSAM improved the DSC score and NSD by 9% and 22.28% compared to the pre-trained LiteMedSAM model as the baseline model, respectively. However, the fine-tuned and EfficientViT models with majority voting yielded little to no accuracy improvement. These results (Figure 3) demonstrate the effectiveness of incorporating 3D spatial information to improve segmentation accuracy without further training.

Given these initial findings, we proposed a solution that includes multiple models and techniques tailored for different imaging modalities rather than having a one-size-fits-all solution. EfficientViT-SAM is applied to microscopy and X-ray images using a bounding box as the prompt. The original LiteMedSAM model is utilized for other modalities, with bounding boxes automatically generated from previous slice segmentation for 3D MR and CT data. To improve segmentation accuracy for PET, a majority voting mechanism is applied to integrate 3D spatial information.

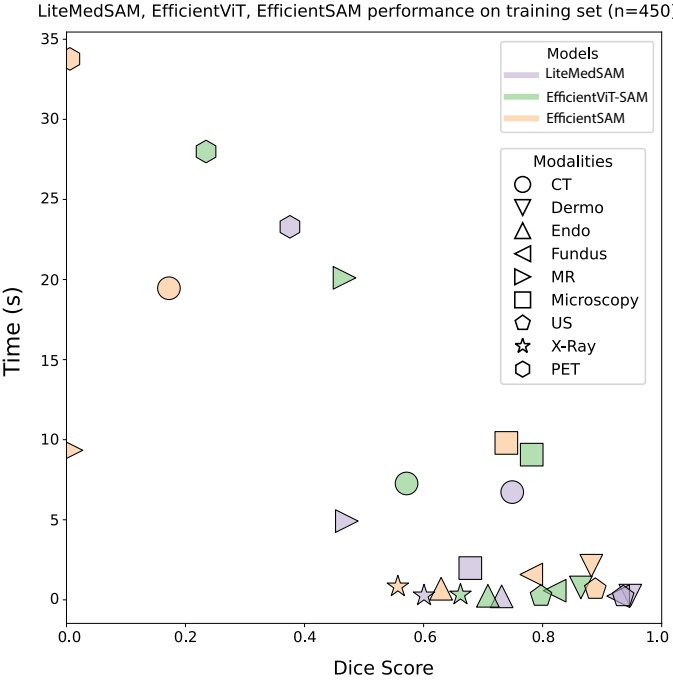

**Fig. 2.** Runtime as a function of DSC score across models—LiteMedSAM, EfficientViT-SAM, and EfficientSAM—and 9 modalities sampled from the training set. All models were converted to OpenVINO graphs for C++ inference pipeline.

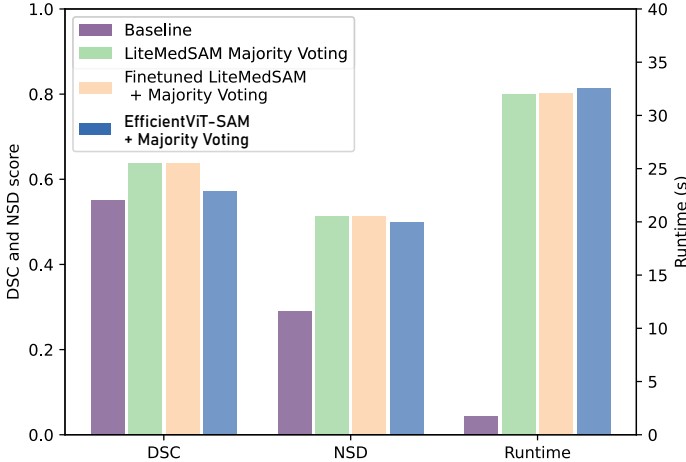

**Fig. 3.** DSC and NSD scores for PET validation data (n = 3) segmentation, as well as the runtime, are shown for three distinct approaches: baseline model, pre-trained LiteMedSAM with majority voting to incorporate 3D spatial information, finetuned LiteMedSAM with majority voting and EfficientViT-SAM with majority voting. The baseline model is in the Pytorch framework; the other three are OpenVINO models in the C++ pipeline.

### 4.1   Quantitative results on validation set

Our proposed method was evaluated in PyTorch, ONNX, and OpenVINO implementation, comparing DSC and NSD scores across all modalities to further understand the impact of different frameworks.

Overall, OpenVINO has a lower average DSC score than PyTorch and ONNX but achieves the highest average NSD score (Table 4). The performance variation among the three formats may be attributed to differences in the preprocessing step. OpenVINO's inference script used OpenCV's bilinear interpolation with fixed coefficients for faster processing, while PyTorch and ONNX employed bilinear interpolation with custom coefficients based on image dimensions.

While OpenVINO showed only marginal differences in accuracy compared to ONNX and PyTorch for 2D images, it exhibited significant discrepancies for 3D images, where bounding boxes were derived from masks in previous slices (Table 4). Its accuracy dropped by approximately 2% for CT and 3% for MR, while gaining notable improvement on PET images, achieving approximately 3% higher DSC and 8% higher NSD scores.

**Table 4.** Accuracy on the validation set using our proposed approach (a combination of models) in Pytorch, ONNX and OpenVINO frameworks.

| Target | DSC (%) | | | NSD (%) | | |
|---|---|---|---|---|---|---|
| | PyTorch | ONNX | OpenVINO | PyTorch | ONNX | OpenVINO |
| CT | 92.19 | 92.19 | 90.05 | 94.71 | 94.74 | 92.66 |
| MR | 88.88 | 88.85 | 85.85 | 92.19 | 92.17 | 89.49 |
| PET | 60.68 | 60.33 | 63.87 | 43.21 | 42.94 | 51.40 |
| US | 94.77 | 94.77 | 94.50 | 96.81 | 96.83 | 96.56 |
| X-Ray | 76.31 | 76.13 | 76.46 | 81.52 | 81.15 | 81.49 |
| Dermotology | 92.47 | 92.41 | 92.2 | 93.86 | 93.80 | 93.58 |
| Endoscopy | 96.04 | 96.05 | 96.07 | 98.11 | 98.12 | 98.16 |
| Fundus | 94.81 | 94.77 | 93.28 | 96.42 | 96.38 | 94.88 |
| Microscopy | 82.79 | 83.03 | 83.25 | 89.67 | 89.58 | 89.96 |
| Average | 86.55 | 86.50 | 86.17 | 87.39 | 87.3 | 87.57 |

### 4.2   Qualitative results on training set

The resulting segmentations for PET images from the training set, obtained using the pre-trained LiteMedSAM model with fixed 3D bounding box prompts and majority voting across anatomical views, are shown in Figure 4. In some instances, the model under-segments, generating a segmentation that is much smaller than ground truth (upper panel in Figure 4; DSC = 0.39, NSD = 0.45). However, the segmentation quality is considerably better in other cases, accurately capturing the target regions (lower panel in Figure 4; DSC = 0.91,

NSD = 0.93). This indicates that while the majority voting mechanism generally improves segmentation accuracy, there are still scenarios where the model's performance could be enhanced, especially for smaller ROIs.

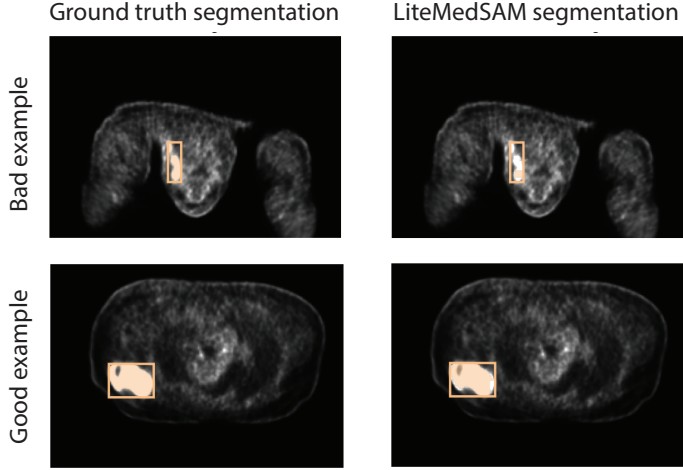

**Fig. 4.** Ground truth (left) and predicted segmentations (right) using majority voting with pre-trained LiteMedSAM model on PET images: `3D_PET_Lesion_PETCT_185da4c8b6` (upper panel) and `3D_PET_Lesion_PETCT_01140d52d8` (lower panel) in the training set.

As for microscopy images, the EfficientViT-SAM model prompted with a bounding box, demonstrated reasonable segmentation performance as in Figure 5. Similarly to LiteMedSAM, in some instances, EfficientViT-SAM underperformed at the segmentation of small ROIs, leading to inaccurate segmentation (upper panel in Figure 5; DSC = 0.53, NSD = 0.71). In contrast, the EfficientViT-SAM model performs exceptionally well for larger targets, delivering highly accurate segmentations (lower panel in Figure 5; DSC = 0.90, NSD = 0.91). This suggests that the model's effectiveness is influenced by the size of the segmentation target, and while it excels with larger regions, additional refinement may be needed for smaller targets.

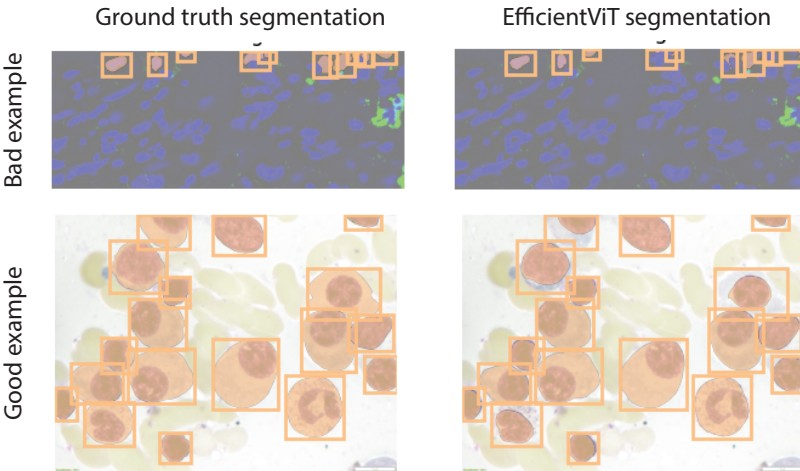

**Fig. 5.** Ground truth (left) and predicted segmentations (right) using EfficientViT-SAM model with bounding box prompt on microscopy images: `2D_Microscope_NeurIPS22CellSeg_cell_00020` (upper panel) and `2D_Microscope_NeurIPS22CellSeg_cell_00020` (lower panel) in the training set.

Overall, our results suggest that the selected strategies exhibit promising performance in their respective modalities.

### 4.3   Segmentation efficiency results on validation set

The runtime measurement starts from loading the Docker image and ends with saving the segmentation. The OpenVINO models in the C++ pipeline consistently outperform the same approach using PyTorch and ONNX framework regarding runtime across all examples (Table 5).

### 4.4   Results on final testing set

Table 6 presents the DSC, NSD, and runtime metrics for our proposed approach using OpenVINO, compared to the baseline model. While the DSC and NSD scores of the proposed approach are marginally lower than the baseline, it offers a significant advantage with respect to runtime, achieving inference speed over three times faster. This highlights a trade-off between segmentation accuracy and computational efficiency, demonstrating that the proposed method substantially improves inference speed with a slight compromise in performance.

### 4.5   Limitation and future work

Despite LiteMedSAM being trained on microscopy data and EfficientViT-SAM having no medical image training data, the zero-shot generalizability of EfficientViT-

**Table 5.** Quantitative evaluation of segmentation efficiency in terms of runtime (s) on 12th Gen Intel® Core™ i7-12700 × 20 @ 2.10GHz RAM 32GB, Docker version 26.0.0. PyTorch is our proposed approach (a combination of models) using PyTorch framework within a Python inference pipeline, ONNX represents the converted graph of our PyTorch approach inferred using a Python script, and OpenVINO is the converted model from ONNX that is inferred using C++ script.

| Case ID | Size | No. Objects | PyTorch | ONNX | OpenVINO |
|---|---|---|---|---|---|
| 3DBox_CT_0566 | (287,512,512) | 6 | 330.18 | 208.05 | 45.46 |
| 3DBox_CT_0888 | (237,512,512) | 6 | 85.81 | 56.96 | 16.75 |
| 3DBox_CT_0860 | (246,512,512) | 1 | 15.33 | 9.46 | 5.24 |
| 3DBox_MR_0621 | (115,400,400) | 6 | 133.41 | 88.25 | 16.57 |
| 3DBox_MR_0121 | (64,290,320) | 6 | 88.01 | 57.26 | 10.59 |
| 3DBox_MR_0179 | (84,512,512) | 1 | 13.86 | 9.14 | 4.52 |
| 3DBox_PET_0001 | (264,200,200) | 1 | 42.54 | 31.72 | 10.23 |
| 2DBox_US_0525 | (256,256,3) | 1 | 3.62 | 2.04 | 1.13 |
| 2DBox_X-Ray_0053 | (320,640,3) | 34 | 3.27 | 2.67 | 1.49 |
| 2DBox_Dermoscopy_0003 | (3024,4032,3) | 1 | 4.03 | 2.33 | 1.52 |
| 2DBox_Endoscopy_0086 | (480,560,3) | 1 | 3.60 | 2.04 | 1.10 |
| 2DBox_Fundus_0003 | (2048,2048,3) | 1 | 3.64 | 2.07 | 1.18 |
| 2DBox_Microscope_0008 | (1536,2040,3) | 19 | 4.63 | 2.52 | 1.49 |
| 2DBox_Microscope_0016 | (1920,2560,3) | 241 | 12.81 | 7.82 | 7.01 |

**Table 6.** Accuracy on the **testing set** using our proposed approach (a combination of models) in the OpenVINO framework.

| Target | DSC (%) | | NSD (%) | | Runtime (s) | |
|---|---|---|---|---|---|---|
| | Baseline | Our solution | Baseline | Our solution | Baseline | Our solution |
| CT | 55.75 | 49.13 | 58.48 | 52.12 | 38.78 | 11.50 |
| MR | 64.80 | 58.80 | 62.75 | 59.07 | 18.57 | 5.55 |
| PET | 76.94 | 71.36 | 66.98 | 60.11 | 14.90 | 12.24 |
| US | 85.24 | 83.54 | 89.73 | 89.30 | 8.96 | 2.37 |
| X-Ray | 85.51 | 78.17 | 94.40 | 88.97 | 9.95 | 1.98 |
| OCT | 73.31 | 67.29 | 80.20 | 73.75 | 8.39 | 1.89 |
| Endoscopy | 94.41 | 94.40 | 96.95 | 96.93 | 7.56 | 1.81 |
| Fundus | 87.47 | 87.49 | 89.58 | 89.57 | 8.77 | 1.91 |
| Microscope | 84.36 | 87.61 | 86.15 | 89.35 | 16.34 | 2.19 |
| **Average** | **78.64** | **75.31** | **80.58** | **77.69** | **14.69** | **4.61** |

SAM for this modality outperformed LiteMedSAM. Considering expensive training costs and the difficulty of collecting medical image data at a large scale, this finding motivates further exploration of zero-shot capabilities of foundation models trained on large-scale natural scene images to segment medical images without further fine-tuning.

In our current implementation of the majority vote mechanism for PET scans, all anatomical views have the same weight. Future work may consider the effect of weight adjustment on the final prediction to understand the contributions of each anatomical view in the overall segmentation improvement. To further optimize runtime for PET scans, one may test reducing the number of anatomical views for inference, for example, from three to two. This may lead to minor segmentation accuracy degradation but substantial runtime gains.

## 5   Conclusion

In conclusion, our solution includes various models customized for distinct imaging modalities: EfficientViT-SAM for microscopy and X-ray; the original LiteMedSAM for other modalities with an automatic bounding box generation mechanism for 3D data and majority voting to integrate 3D spatial information for PET data. Overall, the runtime of OpenVINO with the C++ inference script outperformed the baseline provided by the challenge. While accuracy on Microscopy images surpassed the baseline on the testing set, accuracy for other modalities remains suboptimal and requires further improvement.

**Acknowledgements**  We thank all the data consortiums and researchers involved in data acquisition for making the training medical imaging data publicly available, CodaLab [17] for hosting the challenge platform, and the CVPR 2024 challenge organizers.

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

**Table 7.** Checklist Table. Please fill out this checklist table in the answer column.

| Requirements | Answer |
| --- | --- |
| A meaningful title | Yes |
| The number of authors ($\leq$6) | 6 |
| Author affiliations and ORCID | Yes |
| Corresponding author email is presented | Yes |
| Validation scores are presented in the abstract | Yes |
| Introduction includes at least three parts: background, related work, and motivation | Yes |
| A pipeline/network figure is provided | Figure 1 |
| Pre-processing | Page 3 |
| Strategies to data augmentation | Page 6 |
| Strategies to improve model inference | Page 5 |
| Post-processing | Page 5 |
| Environment setting table is provided | Table 2 |
| Training protocol table is provided | Table 3 |
| Ablation study | Page 7, 8 |
| Efficiency evaluation results are provided | Table 5 |
| Visualized segmentation example is provided | Figure 4 - 5 |
| Limitation and future work are presented | Yes |
| Reference format is consistent. | Yes |
| Main text $>=$ 8 pages (not include references and appendix) | Yes |