# OpenReview forum: "Modality-Specific Strategies for Medical Image Segmentation using Lightweight SAM Architectures"
_thecvf.com/CVPR/2024/Workshop/MedSAMonLaptop — CVPR24 MedSAMonLaptop_

### Official Review · Reviewer_57MJ · 2024-06-10
**It's complete, but it needs a little more.**

**Rating:** 5
**Confidence:** 5

**Review:**

This paper uses an ensemble of different SAM frameworks to iteratively train the modaltily-agnostic framework. The efficiency of the inference phase is optimized by exporting the model to ONNX format for lightweight inference.

Comment:
1. My major concern is the training data. As stated by the authors, they only train and fine-tune both LiteMedSAM and EfficientSAM with only PET and Microscopy images, which is insufficient compared to the whole dataset provided by the challenge.
2. The inference time is confusion. As EfficientSAM used 1024 pixels for input resolution, inference with this large image should result as higher inference time than reported in Table 6, not to mention they are the combination of multiple framework (MedSAM, LiteMedSAM, EfficientSAM)
3. Why the reported results of Pytorch and ONNX is so different, as the ONNX models are quantized, the decrease in results is expected, but here it is higher in some modalities.

---

### Official Review · Reviewer_6oud · 2024-06-10
**Recommend**

**Rating:** 8
**Confidence:** 4

**Review:**

This paper investigates the deployment of lightweight medical image segmentation models on CPUs to improve performance and inference speed. It utilizes models like EfficientSAM and LiteMedSAM for different modality, optimized through methods such as model quantization and the ONNX format. The authors demonstrate significant accuracy and runtime improvements across various medical imaging modalities, including PET scans, by integrating strategies like majority voting and fine-tuning with Sharpness-aware Minimization. The detailed methodology and experimental results indicate that the paper is thorough and provides sufficient information for reproducibility.

As a summary, I recommend this paper for acceptance due to its significant contributions to performance and efficiency on lightweight MedSAM, with good details for reproducing.

---

### Official Review · Reviewer_mrPP · 2024-06-13
**The paper presents an investigation into deploying medical image segmentation models on CPU platforms, focusing on accuracy and runtime efficiency. The study explores different lightweight SAM-based models tailored for specific modalities, such as EfficientSAM, LiteMedSAM, and fine-tuned models, aiming to balance between precision and computational efficiency. The authors propose a variety of techniques, including majority voting for PET scans and model quantization, to optimize performance and deployment feasibility in clinical settings.**

**Rating:** 7
**Confidence:** 3

**Review:**

## Strengths:

### 1. Comprehensive Evaluation:

The paper provides a thorough comparison of multiple models (EfficientSAM, LiteMedSAM, MedSAM) across various medical image modalities, highlighting the strengths and limitations of each model.


### 2. Methodological Rigor:

The incorporation of advanced techniques such as model quantization, ONNX format for runtime optimization, and fine-tuning strategies like Sharpness-aware Minimization demonstrates methodological rigor and a strong understanding of the field.

### 3. Practical Relevance:

By focusing on CPU deployment and evaluating performance in real-world clinical settings, the study has significant practical implications for making advanced segmentation models accessible in resource-constrained environments.
## Weaknesses:

### 1. Efficiency:

While the challenge is to deploy the MedSAM on a laptop, the proposed method's efficiency is only slightly better than that of baseline. (only 10% improvement on case 3DBox_CT_0566).

---

### Official Review · Reviewer_Kbs9 · 2024-06-15
**The authors combined the advantages of EfficientSAM and LiteMedSAM to design a Modality-Specific Strategie, and used the ONNX framework to improve the efficiency of inference.**

**Rating:** 9
**Confidence:** 4

**Review:**

The authors have amalgamated the strengths of EfficientSAM and LiteMedSAM methodologies to formulate Modality-Specific Strategies, leveraging the ONNX framework to enhance inference efficiency. Interestingly, a diverse array of enhancements tailored to various modalities and data dimensions are investigated. While the authors have introduced a multitude of incremental advancements in a single framework, certain pivotal intricacies remain unaddressed, necessitating further validation experiments.

1) In the context of 3D data processing, the authors employ the previous image's mask to generate prompts for subsequent images, capitalizing on the sequential image continuity. However, the segmentation quality of individual slices may occasionally be suboptimal, potentially impacting the subsequent slice's box generation. How does the author mitigate this challenge? Is there a fusion mechanism, perhaps involving a weighted combination, between newly generated and original boxes?

2) The authors experiment with introducing random sampling points as novel prompts alongside box prompts. What structural modifications on models ensue from this inclusion? Is there an adoption of a distinct prompt encoder? How are the training dynamics altered to accommodate both box and point prompts concurrently? What is the number of sampling points? Is the number of sampling points proportional to the box area?

3) Within the framework's majority voting mechanism for PET data, do all multiple views carry identical weighting coefficients? Could incorporating learnable weights potentially enhance performance further? Alternatively, is there merit in evaluating segmentation efficacy on specific views and adjusting weights through statistical methodologies for optimization?

---

### Official Review · Reviewer_v9sL · 2024-06-16
**Detailed Enhancement Recommendations for Methodology, Experiments, and Results Sections**

**Rating:** 4
**Confidence:** 3

**Review:**

### Methodology:
- **Section 2.1 and 2.2**: These sections describe the preprocessing and the proposed method, but a more detailed algorithm description and pseudocode are needed for the reader to better understand the specific steps of the proposed method.
- In the quantization process, it is necessary to detail the basis for the choice of quantization strategy, as well as how to balance accuracy and efficiency.

### Experiments:
- **Section 3.1**: The datasets and samplers mentioned need a more detailed description, including the source, diversity, and scale of the datasets.
- **Section 3.2**: The evaluation metrics and loss functions used need a deeper explanation, including why these specific metrics and functions were chosen.
- **Section 3.3**: The training protocol needs a more detailed parameter setting and adjustment strategy.

### Results:
- **Section 4.1**: The inference speed results need to be compared with existing technologies to show the advantages of the proposed method.
- **Section 4.2**: The quantitative results need a deeper analysis, including the performance differences of the model on different modalities and possible reasons.
- **Section 4.3**: The qualitative results need more case studies to show the performance of the model in different situations.

---

### Decision · Program_Chairs · 2024-10-01

Accept